# Undecidability in quantum thermalization

Naoto Shiraishi [1✉] & Keiji Matsumoto[2✉]

The investigation of thermalization in isolated quantum many-body systems has a long history, dating back to the time of developing statistical mechanics. Most quantum many-body systems in nature are considered to thermalize, while some never achieve thermal equilibrium. The central problem is to clarify whether a given system thermalizes, which has been addressed previously, but not resolved. Here, we show that this problem is undecidable. The resulting undecidability even applies when the system is restricted to one-dimensional shift-invariant systems with nearest-neighbour interaction, and the initial state is a fixed product state. We construct a family of Hamiltonians encoding dynamics of a reversible universal Turing machine, where the fate of a relaxation process changes considerably depending on whether the Turing machine halts. Our result indicates that there is no general theorem, algorithm, or systematic procedure determining the presence or absence of thermalization in any given Hamiltonian.

[1] Department of Physics, Gakushuin Univerisity, Toshima-ku, Tokyo, Japan. [2] Quantum Computation Group, National Institute of Informatics, Chiyoda-ku, Tokyo, Japan. ✉email: naoto.shiraishi@gakushuin.ac.jp; keiji@nii.ac.jp

Thermalization, or relaxation to equilibrium, in isolated quantum many-body systems is a ubiquitous yet profound phenomenon. The history of investigation of thermalization dates back to Boltzmann[1] and von Neumann[2], and many theoretical physicists have studied this problem. The problem originated in the field of nonequilibrium statistical mechanics. However, some techniques developed in quantum information theory have gained attention to provide fresh insight into this old problem[3]. From the experimental side, the recent development of experimental techniques to manipulate cold atoms enabled us to observe thermalization of isolated quantum many-body systems in the laboratory[4–9]. Experimentalists not only tested established theoretical results, but also revealed some unexpected behaviours[9].

A central problem in this field is whether a given system thermalizes[3,10]. Although almost all-natural quantum many-body systems are expected to thermalize, some systems, including integrable and localized systems, are known to never achieve thermalization[11–15]. To resolve this problem, the eigenstate thermalization hypothesis (ETH) has been raised as a clue to understanding thermalization phenomena. The ETH claims that all the energy eigenstates of a given Hamiltonian are thermal, that is, indistinguishable from the equilibrium state, as long as we observe macroscopic observables[16–21]. Studies based on numerical simulations support that most non-integrable thermalizing systems satisfy the ETH[20,22–24]. In contrast, recent theoretical studies and elaborated experiments have revealed that some non-integrable systems do not satisfy the ETH[9,25–31]. Numerous other theoretical ideas, including largeness of effective dimension[10], typicality[10,32–34], and quantum correlation[35–37] have been proposed to elucidate thermalization phenomena; however, none of them provides a decisive answer.

We approach the problem of thermalization from the opposite side. We examine the difficulty of the problem from the viewpoint of theoretical computer science. This type of approach is employed in some problems in physics, including prediction of dynamical systems[38], repeated quantum measurements[39], and the spectral gap problem[40]. In this approach, these problems were unexpectedly shown to be undecidable, that is, there is no algorithm to determine, e.g., the presence or absence of a spectral gap in arbitrary systems in the case of the spectral gap problem.

Our main achievement in this paper is the finding that whether a given system thermalizes or not with respect to a given observable is undecidable in general. This result shows not merely the difficulty of this problem, but also the logical impossibility of solving it. Hence, the fate of thermalization in a general setup is independent of the basic axioms of mathematics, as implied in the Gödel's incompleteness theorem[41]. We prove this by demonstrating that the relaxation and thermalization phenomena in one-dimensional systems have the power of universal computation. Our result not only sets a limit on what we can know about quantum thermalization, but also elucidates a rich variety of thermalization phenomena, which can implement any computational task.

## Results

**Statement of main results**. We first clarify the precise statements of our results, namely, the undecidability of relaxation and thermalization. Since the undecidability of thermalization can be obtained by modifying the result on relaxation, we shall mainly treat relaxation and briefly comment on how to extend this result to thermalization. Throughout this study, we consider a one-dimensional lattice system of size $L$ with the periodic boundary condition (we finally take $L \to \infty$ limit), with $d$-dimensional local Hilbert space $\mathcal{H}$. Although we do not specify the necessary dimension, we roughly estimate that $d \simeq 120$ suffices to obtain undecidability, which is minuscule

compared to other results of undecidability in physics[40]. Let $|\psi(t)\rangle$ be the state of the system at time $t$. The long-time average of an observable $\mathcal{A}_L$ for a given initial state $|\psi(0)\rangle = |\psi_0^L\rangle$ is given by $\bar{\mathcal{A}}_L = \lim_{T \to \infty} \frac{1}{T} \int_0^T dt \langle \psi(t)| \mathcal{A}_L |\psi(t)\rangle$. Our interest takes the form of whether the thermodynamic limit of the long-time average $\bar{\mathcal{A}}_L$, denoted by $\bar{\mathcal{A}} := \lim_{L \to \infty} \bar{\mathcal{A}}_L$, converges to the vicinity of a given target value $A^*$. This question concerns the fate of a relaxation process with an initial state, an observable, and a Hamiltonian. If $A^*$ is equal to the equilibrium value $\mathcal{A}^{MC} := \lim_{L \to \infty} \mathrm{Tr}[\mathcal{A}_L \rho_L^{MC}]$ with the microcanonical state $\rho_L^{MC}$, this question asks whether thermalization with respect to $\mathcal{A}$ takes place. We remark that we take the long-time limit ($T \to \infty$) first, and then take the thermodynamic limit ($L \to \infty$). The symbol $\mathcal{A}$ means the thermodynamic limit of $\mathcal{A}_L$, while the order of the limit is always in the aforementioned one.

We restrict the class of the Hamiltonians, observables, and initial states to simple ones. The Hamiltonian of the system is restricted to be nearest-neighbour interaction and shift-invariant. Hence, the $d^2 \times d^2$ local Hamiltonian $h_{i,i+1}$, which acts only on sites $i$ and $i + 1$, fully determines the system Hamiltonian as $H := \sum_i h_{i,i+1}$. We further restrict observables to a spatial average of a single-site operator: $\mathcal{A}_L := \frac{1}{L} \sum_{i=1}^L A_i$, where $A_i$ acts only on the site $i$. In addition, we restrict the initial state as the following form of a product state: $|\psi_0^L\rangle = |\phi_0\rangle \otimes |\phi_1\rangle \otimes |\phi_1\rangle \otimes \cdots \otimes |\phi_1\rangle$, where $|\phi_0\rangle$ and $|\phi_1\rangle$ are states on a single-site orthogonal to each other; $\langle \phi_0 | \phi_1 \rangle = 0$.

In our setup, both the observable ($A$) and the initial state ($|\phi_0\rangle$ and $|\phi_1\rangle$) are given arbitrarily and fixed. Moreover, we put a promise that either $|\bar{\mathcal{A}} - A^*| < \varepsilon_1$ or $|\bar{\mathcal{A}} - A^*| > \varepsilon_2$ holds with errors $0 < \varepsilon_1 < \varepsilon_2$. An alternative expression of the above promise is that we are allowed to answer incorrectly for $\varepsilon_1 \leq |\bar{\mathcal{A}} - A^*| \leq \varepsilon_2$. The ratio of errors $M := \varepsilon_2/\varepsilon_1$ can be set arbitrarily large. The input of this decision problem is the Hamiltonian. Even in this very simple setup, we show that whether the long-time average of $\mathcal{A}$ from this initial state $|\psi_0^L\rangle$ under a given Hamiltonian $H$ relaxes to the vicinity of a given value $A^*$ is undecidable.

**Theorem 1** *Given two states $|\phi_0\rangle$ and $|\phi_1\rangle$ on a single site, orthogonal to each other, and a single-site operator $A$ arbitrarily. We require that there exists a state $|\phi_2\rangle$ orthogonal to $|\phi_0\rangle$ and $|\phi_1\rangle$ such that $\langle \phi_2|A|\phi_2\rangle \neq \langle \phi_1|A|\phi_1\rangle$. The initial state and the observable are set as $|\psi_0^L\rangle = |\phi_0\rangle \otimes |\phi_1\rangle \otimes \cdots \otimes |\phi_1\rangle$ and $\mathcal{A}_L := \frac{1}{L} \sum_{i=1}^L A_i$. Here, the long-time average $\bar{\mathcal{A}}$ is a function of the Hamiltonian $H$. We also fix $M > 0$ arbitrarily large. Then, there exist $\varepsilon_1, \varepsilon_2$ with $\varepsilon_2 = M\varepsilon_1$, and $A^*$ which satisfy the following: we suppose the promise that either $|\bar{\mathcal{A}} - A^*| < \varepsilon_1$ or $|\bar{\mathcal{A}} - A^*| > \varepsilon_2$ holds (see Fig. 1). In this setting, deciding which is true for a given shift-invariant nearest-neighbour interaction Hamiltonian $H = \Sigma_i h_{i,i+1}$ is undecidable.*

If $A^*$ is equal to the equilibrium value $\mathcal{A}^{MC}$, our result reads undecidability of thermalization: Whether a given system with a fixed initial state thermalizes or not with respect to a fixed observable $A$ is undecidable. By defining $A_{max}^{01} := \max |\psi\rangle \in \mathrm{span}\{|\phi_0\rangle, |\phi_1\rangle\} \langle \psi|A|\psi\rangle$ and $A_{min}^{01} := \min_{|\psi\rangle \in \mathrm{span}\{|\phi_0\rangle, |\phi_1\rangle\}} \langle \psi|A|\psi\rangle$, the precise statement can be expressed as follows:

**Theorem 2** *Given two states $|\phi_0\rangle$ and $|\phi_1\rangle$ on a single site, orthogonal to each other, and a single-site operator $A$ arbitrarily. We require that there exist states $|\phi_2\rangle$ and $|\phi_3\rangle$ orthogonal to $|\phi_0\rangle$, $|\phi_1\rangle$, $A|\phi_0\rangle$, and $A|\phi_1\rangle$ such that $\langle \phi_2|A|\phi_2\rangle > A_{max}^{01}$ and $\langle \phi_3|A|\phi_3\rangle < A_{min}^{01}$. The initial state and the observable are set as*

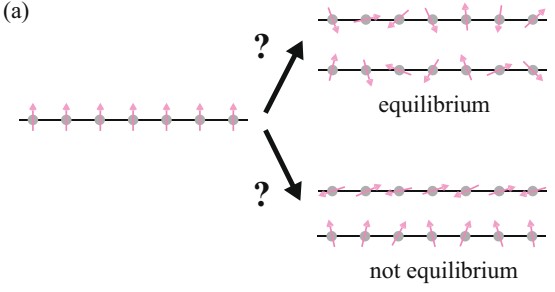

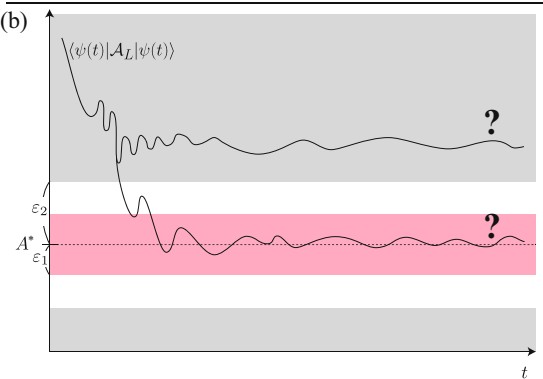

**Fig. 1 The problem of thermalization concerns the long-time average of the observable. a** We consider whether a nonequilibrium initial state relaxes to the equilibrium or not. **b** More precisely, we decide whether the long-time average of $\langle\psi(t)|\mathcal{A}_L|\psi(t)\rangle$ converges to the value $A^*$ with precision $\varepsilon_1$, or deviates from $A^*$ at least $\varepsilon_2 > \varepsilon_1$ in the thermodynamic limit (If the long-time average settles between $\varepsilon_1$ and $\varepsilon_2$, we do not have to answer). This problem is shown to be an undecidable problem.

$|\psi_0^L\rangle = |\phi_0\rangle \otimes |\phi_1\rangle \otimes \cdots \otimes |\phi_1\rangle$ and $\mathcal{A}_L := \frac{1}{L}\sum_{i=1}^L A_i$. We also fix $M > 0$ arbitrarily large. Then, there exist $\varepsilon_1$, $\varepsilon_2$ with $\varepsilon_2 = M\varepsilon_1$, which satisfy the following: We suppose the promise that either $|\bar{\mathcal{A}} - \mathcal{A}^{\mathrm{MC}}| < \varepsilon_1$ or $|\bar{\mathcal{A}} - \mathcal{A}^{\mathrm{MC}}| > \varepsilon_2$ holds. In this setting, deciding which is true for a given shift-invariant nearest-neighbour interaction Hamiltonian $H = \Sigma_i h_{i,i+1}$ is undecidable.

The condition on the presence of $|\phi_2\rangle$ and $|\phi_3\rangle$ ensures that the initial state is not at the edge of the spectrum of $A$. We note that the equilibrium value $\mathcal{A}^{\mathrm{MC}}$ depends on the choice of the Hamiltonian, and thus the promise restricts the class of Hamiltonians.

**Mapping classical Turing machines to a quantum system.** We here sketch the main idea of the proof. A rigorous proof is presented in the Supplementary Note. We first introduce a key ingredient, the halting problem of a Turing machine (TM), which is a prominent example of undecidable problems. The halting problem of a TM asks whether the TM with a given input halts at some time or does not halt and runs forever. Turing proved in his celebrated paper that there exists no general procedure to solve the halting problem[42].

Following various studies demonstrating undecidability[43], we apply the reduction to the halting problem. We shall construct a family of Hamiltonians with which the long-time average of an observable is connected to the halting or non-halting of a TM. Below, a universal reversible Turing machine (URTM) is arbitrarily given and fixed, whose possible input code is denoted by $\boldsymbol{u}$.

**Lemma:** *Given a complete orthogonal normal basis of the local Hilbert space $\{|e_i\rangle\}$ and an observable $A$ on a single site satisfying $\langle e_1|A|e_1\rangle = 0$ and $\langle e_2|A|e_2\rangle > 0$ arbitrarily. Then, for any $\eta > 0$,*

*there exists a shift-invariant nearest-neighbour interaction Hamiltonian H and a set of unitary operators $\{V_{\boldsymbol{u}}\}$ on the local Hilbert space $\mathcal{H}$ corresponding to all possible inputs for the fixed URTM $\boldsymbol{u}$ such that they satisfy $V_{\boldsymbol{u}}|e_0\rangle = |e_0\rangle$ for any $\boldsymbol{u}$ and the following property:*
*Set the initial state as*

$$|\psi_0^L\rangle = (V_{\boldsymbol{u}}|e_0\rangle) \otimes (V_{\boldsymbol{u}}|e_1\rangle)^{\otimes L-1}. \tag{1}$$

*If the URTM halts with the input $\boldsymbol{u}$, then*

$$\bar{\mathcal{A}} \geq \left(\frac{1}{4} - \eta\right)\langle e_2|A|e_2\rangle \tag{2}$$

*holds, and if the URTM does not halt with the input $\boldsymbol{u}$, then*

$$\bar{\mathcal{A}} \leq \eta \tag{3}$$

*holds.*

By setting the initial state, the observable, and the Hamiltonian in Theorem 1 as $|e_0\rangle \otimes (|e_1\rangle)^{\otimes L-1}$, $V^\dagger A V$, and $V_{\boldsymbol{u}}^{\dagger \otimes L} H V_{\boldsymbol{u}}^{\otimes L}$, respectively, the degree of freedom in the choice of unitary transformation is mapped onto that of the local Hamiltonian. Then, the setup of Lemma can be mapped onto that of Theorem 1 by shifting the origin of $A$ so that $\langle\phi_1|A|\phi_1\rangle = 0$, and setting $|\phi_i\rangle$ $(i = 0, 1, 2)$ to $|e_i\rangle$. Because the halting problem of the URTM is undecidable, the above lemma directly implies the undecidability of the long-time average in quantum many-body systems.

To prove the lemma, we first introduce an elaborated classical machine that simulates the given URTM and changes the value of $A$ depending on whether the URTM halts. We then construct a quantum many-body system emulating the above classical machine. Since the dynamics of the quantum system is a superposition of classical machines with different inputs, we first compute the long-time average for computational basis initial states, which corresponds to a single input, and then treat the quantum superposition.

**Classical machines.** Here, we outline the construction of a classical TM, which simulates the halting problem of a given URTM and changes the long-time average of the observable $A$ depending on the behaviour of the URTM. This machine consists of three TMs, TM1, TM2, and TM3, on two types of cells, M-cells and A-cells. Unlike conventional TMs, the finite control settles in the line of cells. TM2 simulates the URTM with the input code $\boldsymbol{u}$, whose reversibility is induced by the unique direction property[44]. TM1 decodes the input code $\boldsymbol{u}$ from a sequence of two qubits. Two TMs, TM1 and TM2, work in M-cells. TM3 is a simple TM, which flips the state of A-cells if and only if TM2 halts. Through the above trick, the long-time average $\bar{\mathcal{A}}$ in our system reflects the result of the halting problem of TM2.

An M-cell consists of three layers: The first layer simulates the URTM, and the second and the third layers, both of which consist of sequences of qubits, store the input code of TM2. The relative frequency of 1 in the second layer is set to $\beta$ whose binary expansion is equal to the input code $\boldsymbol{u}$. TM1 decodes a bit sequence $\boldsymbol{u}$ on the first layer from the second and the third layers by estimating the relative frequency of 1 (see the first part of Fig. 2), and then TM2 runs with this input $\boldsymbol{u}$. Throughout this procedure, the machine passes all A-cells transparently.

A-cells are responsible for changing the long-time average of $\mathcal{A}_L$. At the initial state, all A-cells are set to the state $a_1$, whose expectation value of $A$ is zero. If and only if TM2 halts, TM3 starts flipping states of A-cells from $a_1$ to another state $a_2$, whose expectation value of $A$ is a nonzero value. To inflate the

## First part (only M-cells)

TM1

first layer | 1 | 0 | $q_j$ | 0 | ⋯ | 0 | 0 | 0 | 0 | ⋯

Ratio of 1 is $\beta$ (=input $\boldsymbol{u}$).

second layer | 1 | 1 | 0 | 1 | ⋯ | 0 | 1 | 1 | 0 | ⋯

third layer | 0 | 0 | 0 | 0 | ⋯ | 0 | 0 | 1 | 0 | ⋯

Read qubits up to here. ←

## Second part

TM2 (with input $\boldsymbol{u}$)

$a_1$ | $q_u$ | 0 | 1 | ⋯ | 0 | $a_1$ | $a_1$ | 0 | ⋯

↓ If TM2 halts...

TM3

$a_1$ | 0 | 0 | 1 | ⋯ | 0 | $a_2$ | $a_2$ | $r$ | ⋯ →

flipping $a_1$ to $a_2$

**Fig. 2 Roles of three layers in M-cells and schematic of dynamics of two Turing machines.** [Top]: In the first part, a Turing machine, TM1, decodes a bit sequence $\boldsymbol{u}$ on the first layer through the estimation of the number of $|1\rangle$ in the second layer (step (i)). The relative frequency of 1 in the second layer is set to $\beta$, whose binary expansion is equal to $\boldsymbol{u}$. The number of qubits TM1 should read is determined by the leftmost cell with 1 in the third layer. Here, we draw only M-cells and omit A-cells for visibility. [Bottom]: In the second part, a universal reversible Turing machine, TM2, runs with the input $\boldsymbol{u}$ (step (ii)). If TM2 halts, then TM3 starts to flip the state in the A-cells from $a_1$ to $a_2$ (step (iii)). If TM2 does not halt, the states in A-cells are not flipped. Note that we have not drawn the second and third layers of M-cells for visibility. In these figures, $q_j$, $q_u$, and $r$ are examples of internal states of TM1, TM2, and TM3, respectively.

difference between the halting and non-halting cases, we set the initial state such that most of the cells are A-cells.

The procedure is summarized as follows:

(i) TM1 decodes the input code $\boldsymbol{u}$ on the first layer.
(ii) TM2, a URTM, runs with the input $\boldsymbol{u}$ in the first layer.
(iii) If and only if TM2 halts, then TM3 starts flipping the states in A-cells (see the second part of Fig. 2). This induces a visible difference between the long-time average of $\mathcal{A}_L$ in the case of halting and non-halting.
(iv) In the case of halting, the head returns to the cell where TM2 halts due to the periodic boundary condition. By this time, all A-cells have already been flipped, and TM3 stops.

Because the halting problem of TM2 with an arbitrary input $\boldsymbol{u}$ is undecidable, the long-time average of $\mathcal{A}_L$ with an arbitrary local unitary transformation $V_{\boldsymbol{u}}$ is likewise undecidable.

**Hamiltonian construction and its eigenstates**. Our implementation of the classical TM in quantum systems stems from the construction of the Feynman-Kitaev Hamiltonian[45,46], while we delete the clock part. Each site takes one of the states of the finite control or that of a single cell in the tape, or some additional

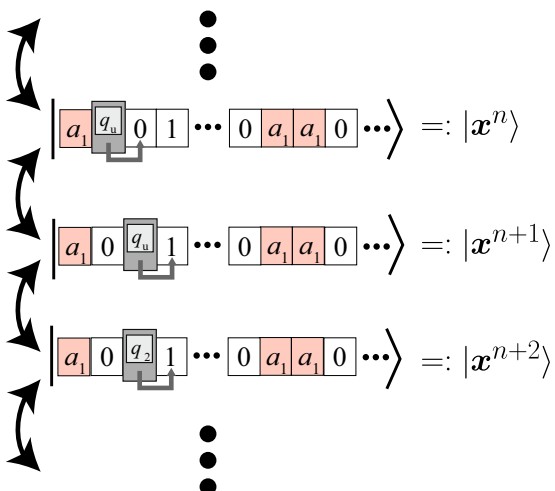

**Fig. 3 Evolution of the quantum state of the total system.** We draw a possible hopping between quantum states in the computational basis. Although here, we depict only the first and second layers for visibility, the quantum state actually consists of three layers. Similar to the Feynman-Kitaev Hamiltonian case, the total Hamiltonian induces the forward and backward one-step time evolution of the TM. We denote the state of the classical TM at the $n$th step by $\boldsymbol{x}^n$. The symbols $a_1$, 0 and 1 represent the states of cells, and $q_u$ and $q_2$ represent the internal states of the TM.

symbols. If the site $i$ is a state of the finite control, then the head reads the site $i+1$ or $i-1$ (see Fig. 3). In the initial state, we set the finite control at site 1, and set all other sites not to the states of the finite control. Because the dynamics conserve the number of sites of the finite control, only a single site takes the state of the finite control at all times.

The dynamics of TM are encoded in the local Hamiltonian as follows. Suppose, e.g., that the cell at the head is $s_a$, the state of the finite control is $q_b$, and the TM moves to the right with keeping the state of the finite control and the cell. Then, the local Hamiltonian $h_{i,i+1}$ must contain the term

$$|s_a, q_b\rangle\langle q_b, s_a| + \text{c.c.} . \tag{4}$$

Similarly, we add all transition rules of TMs (both TM1, TM2, and TM3) to the local Hamiltonian in the form of (4). Owing to the deterministic property of TMs, all the legal states of the total system have a unique descendant state.

Because treatment of almost uniform initial states is slightly complicated, we first take an analogous and easier setting. Our original setting is discussed in the next section. We set the initial state as a non-uniform computational basis state, such that the dynamic of TMs is uniquely determined without quantum fluctuation. Let $|\boldsymbol{x}^1\rangle$ denote the initial configuration of the total system, and $|\boldsymbol{x}^n\rangle$ be the $n$-th state (i.e., after $n-1$ steps from $|\boldsymbol{x}^1\rangle$). By restricting the Hilbert space to the subspace spanned by $\{|\boldsymbol{x}^n\rangle\}$, the total Hamiltonian is expressed as (see also Fig. 3)

$$H' = \sum_{n=1}^{J-1} |\boldsymbol{x}^{n+1}\rangle\langle \boldsymbol{x}^n| + \text{c.c.}, \tag{5}$$

where the $J$-th state is the final state of this dynamics. This Hamiltonian takes the same form as a single-particle system on a closed one-dimensional lattice with only hopping terms. Employing the result on a tridiagonal matrix, eigenenergies and energy eigenstates are calculated as

$$E_j = 2\cos\left(\frac{j\pi}{J+1}\right), \tag{6}$$

$$\left|E_j\right\rangle = \sqrt{\frac{2}{J+1}} \sum_{k=1}^{J} \sin\left(\frac{kj\pi}{J+2}\right)\left|\boldsymbol{x}^k\right\rangle, \tag{7}$$

with $j = 1, 2, \ldots, J$.

By expanding the initial state as $\left|\boldsymbol{x}^1\right\rangle = \sum_{j=1}^{J} c_j|E_j\rangle$, the long-time average of $\mathcal{A}_L$ reads $\bar{\mathcal{A}}_L = \sum_{j=1}^{J}|c_j|^2\langle E_j|A|E_j\rangle$, because all the off-diagonal elements vanish in the long-time average. Since the number of steps until TM2 halts is independent of the system size $L$, by setting $L$ to be sufficiently large, we can make the flipping of A-cells start before $J/2$ steps. In this condition, half of the A-cells have been flipped before $3J/4$ steps, which confirms the nonzero expectation value of $\bar{\mathcal{A}}_L$ in the case of halting. In contrast, in the case of non-halting, the flipping by TM3 does not occur, and hence the long-time average $\bar{\mathcal{A}}_L$ is kept close to zero.

**Uniform initial state**. We now describe the decoding process from the second and third layers of M-cells in our original setting, almost uniform initial states. The sites in the second and third layers are set to $\sqrt{\beta}|1\rangle + \sqrt{1-\beta}|0\rangle$ and $\sqrt{\gamma}|1\rangle + \sqrt{1-\gamma}|0\rangle$, respectively. The state on $m$ of M-cells is a superposition of $2^m \times 2^m$ computational basis states. TM1 runs on each computational basis state, and thus the dynamics of TMs is also a superposition of $2^m \times 2^m$ branches.

The quantity $\beta$ stores the input code in the form such that the binary expansion of $\beta$ equals the input code $\boldsymbol{u}$. TM1 calculates $\beta$ by estimating the relative frequency of the state $|1\rangle$ in the second layer. Due to the law of large numbers, the set of computational basis states such that the relative frequency of $|1\rangle$ is not close to $\beta$ has negligibly small probability amplitude. The quantity $\gamma$ (more precisely, $-1/\ln\gamma$) characterizes the length of qubits that TM1 must read in. TM1 reads the qubits in the second layer until it first encounters $|1\rangle$ in the third layer (the first part of Fig. 2). By setting $\gamma$ to be sufficiently small, the probability of two unwanted cases, namely, (a) TM1 stops decoding before $\boldsymbol{u}$ is decoded to the last, and (b) TM1 can access only an insufficiently small number of qubits in the second layer and fails to estimate the correct $\beta$, becomes negligible.

**From relaxation to thermalization**. We shall sketch how Theorem 2, the undecidability of thermalization, is derived from the proof techniques of Theorem 1. Careful calculation with slightly modified version of TM3 implies that the long-time average $\bar{\mathcal{A}}$ when TM2 halts approaches $\langle e_2|A|e_2\rangle$. Since the basis $\{|e_i\rangle\}$ with $i \geq 2$ can be set arbitrarily in the proof of Theorem 1, it suffices to show the presence of an orthogonal normal basis $\{|e_i\rangle\}$ such that $\langle e_2|A|e_2\rangle = \mathcal{A}^{\mathrm{MC}}$. We remark that the Hamiltonian depends on the basis $\{|e_i\rangle\}$, and thus $\mathcal{A}^{\mathrm{MC}}$ also depends on it through the Hamiltonian.

Since $|\phi_0\rangle$ and $|\phi_1\rangle$ are not at the edge of the spectrum of $A$, the equilibrium value $\mathcal{A}^{\mathrm{MC}}$ always settles between the maximum and the minimum expectation values of $A$ in the subspace orthogonal to $|\phi_0\rangle$, $|\phi_1\rangle$, $A|\phi_0\rangle$, and $A|\phi_1\rangle$. Let $|\sigma_{\max}\rangle$ and $|\sigma_{\min}\rangle$ be states in this subspace accompanying the maximum and minimum expectation values of $A$. We set $|e_2(p)\rangle := \sqrt{p}|\sigma_{\max}\rangle + \sqrt{1-p}|\sigma_{\min}\rangle$ and change $p$ from $p = 0$ to $p = 1$. With recalling $\langle e_2(0)|A|e_2(0)\rangle \leq \mathcal{A}^{\mathrm{MC}} \leq \langle e_2(1)|A|e_2(1)\rangle$ and the continuity of $\langle e_2(p)|A|e_2(p)\rangle$, we find that there exists a proper $p$ (a proper $|e_2\rangle$) which realizes $\langle e_2(p)|A|e_2(p)\rangle = \mathcal{A}^{\mathrm{MC}}$. Using this Hamiltonian with this basis $\{|e_i\rangle\}$, we arrive at the undecidability of thermalization by following the same argument to that of relaxation.

We here remark two points. First, the tuning of $|e_2\rangle$ can be accomplished in the choice of the local Hamiltonian, and both the observable and the initial state are kept as arbitrary fixed parameters. Second, since a finite error from the equilibrium value is allowed, we can compute a proper $p$ (i.e., a proper local Hamiltonian) within this error in a finite number of steps.

**No sufficiently large system size**. Our result claims that we cannot solve the problem of thermalization by any elaborated method even with unlimited computational resource. In order to elucidate the significance of the constructed systems, we compare them with near-integrable systems, $H = H_{\mathrm{int}} + \varepsilon V$, where $H_{\mathrm{int}}$ is an integrable Hamiltonian and $\varepsilon$ is a small parameter. In near-integrable systems, the small parameter $\varepsilon$ determines the necessary system size and time length to distinguish the true thermodynamic limit from prethermal plateaus, and by taking $\varepsilon \to 0$ the necessary size diverges. If our computational resource is unlimited, by setting the system size and running time sufficiently large depending on $\varepsilon$ as determined above we safely obtain the true long-time behaviour in the thermodynamic limit within an arbitrarily small error.

In contrast to near-integrable systems, the constructed systems of undecidability have no such small parameters and no sufficiently large system size. This fact is clearly demonstrated by introducing the *busy beaver function* $\mathrm{BB}(n)$. The busy beaver function gives the largest number of steps which a halting TM with $n$ internal states and an empty input can take. Since the number of internal states can be connected to the length of the input code to a TM with a fixed number of internal states, the busy beaver function also serves as the indicator of the necessary time steps with respect to the length of the input. In terms of thermalization, the busy beaver function provides the necessary system size and time length to observe the true thermodynamic limit. However, the busy beaver function is proven to be uncomputable. More surprisingly, if the Zermelo-Fraenkel set theory with the axiom of choice (ZFC), which is roughly equivalent to the whole of our mathematics, is consistent, then $\mathrm{BB}(748)$ is shown to be uncomputable[47,48]. Notice that all possible TMs with 748 internal states can be implemented by a (large but) finite set of Hamiltonians. These Hamiltonians obviously have no small parameters going to zero, because no quantity tends to go to zero in a finite set. In spite of this, we do not have a sufficiently large system size for these (finite number of) Hamiltonians.

## Discussion

The presence or absence of thermalization in a given quantum many-body system, which has been a topic of debate among researchers in various fields, is proven to be undecidable. Hence, there exists no general systematic procedure to determine the long-time behaviour of quantum many-body systems. The undecidability is still valid for a class of simple systems; one-dimensional systems with a shift-invariant and nearest-neighbour interaction. Our result leads to a fundamental limitation to reach a general theory on thermalization.

Our proof also shows the computational universality of thermalization phenomena. Contrary to the apparent simplicity of thermalization phenomena, the above fact leads to an astonishing consequence that the variety of thermalization phenomena is no less than all possible tasks computers can manage. A striking example bridging physics and mathematics is a system that thermalizes if and only if the Riemann hypothesis is true. The above system reflects the existence of a TM which halts if and only if the Riemann hypothesis is false[49].

From the context of physics, the extremely slow relaxation of our model in case of halting is induced by quasi-conserved non-local quantities, which are close to conserved quantities but not conserved. Recently, some non-integrable systems (the transverse Ising model with z magnetic field) have been reported to relax very slowly, which is caused by quasi-conserved local quantities[23,50,51]. Numerical simulations with ordinal size and time length fail to address thermalization in these systems. Similar things can also be seen in glassy systems, whose connection with computational hardness is also discussed intensively[52]. The extremely slow relaxation in our system might be understood from the aforementioned more general viewpoints, which is worth further investigation.

We remark that our definition of thermalization is conditional with respect to an observable. There exists another definition of thermalization in an unconditional form, where a system is said to thermalize if and only if the system thermalizes with respect to all macroscopic observables. In this article, we do not employ this alternative definition because no shift-invariant system is proven to thermalize in this sense. To prove undecidability, we should prepare infinitely many thermalizing and non-thermalizing systems with proof. Constructing a thermalizing system in this sense is considered to be a very hard problem, and therefore we give up adopting this definition.

We finally comment on the limitations of our result and conclude this study. First, our result does not exclude the possibility that one proves the presence or absence of thermalization in specific systems. Our result only excludes the possibility to obtain a general and ultimate criterion to judge the presence or absence of thermalization. We emphasize that our results do not tarnish the meaningfulness of numerical simulations in ordinal systems with finite size. Second, our undecidability is shown in only a highly artificial model with a particular form of Hamiltonians, which is another limitation of our result. One needs to proceed to a more natural model exhibiting undecidability, or to find a set of a restricted class of physical Hamiltonians whose fate of thermalization is now decidable. These problems are left for future works.

## Method

### Decoding from the second and third layers

We discuss how to decode the input code $u$ from the sequence of two qubits in the second and third layers. The amount of $\beta$, whose binary expansion is equal to $u$, is guessed by the relative frequency of 1's in the second layer (see Fig. 2). We expand $m$ copies of $\sqrt{\beta}|1\rangle + \sqrt{1-\beta}|0\rangle$ as

$$\left(\sqrt{\beta}|1\rangle + \sqrt{1-\beta}|0\rangle\right)^{\otimes m} = \sum_{w \in \{0,1\}^{\otimes m}} \sqrt{\beta}^{N_1(w)} \sqrt{1-\beta}^{m-N_1(w)} |w\rangle, \quad (8)$$

where $w$ is a sequence of 01 with length $m$, and $N_1(w)$ is the number of 1's in the binary sequence $w$. The probability amplitude for a state $|w\rangle$ is $|c_w|^2 = \beta^{N_1(w)}(1-\beta)^{m-N_1(w)}$. Due to the law of large numbers, the probability amplitude for states with relative frequency of 1's close to $\beta$ converges to 1 in the large $m$ limit:

$$\lim_{m\to\infty} \sum_{w: \frac{N_1(w)}{m} \simeq \beta} |c_w|^2 = 1. \quad (9)$$

Here, the symbol $\frac{N_1(w)}{m} \simeq \beta$ means that $\frac{N_1(w)}{m}$ is close to $\beta$, whose rigorous definition is presented soon later (in (11)). Hence, if $m$ is sufficiently large compared with the length of the input code, TM1 guesses $\beta$ correctly from the frequency of 1's.

The length $m$ is determined by another bit sequence, $\sqrt{\gamma}|1\rangle + \sqrt{1-\gamma}|0\rangle$, in the third layer. Let $0 < \xi < 1$ be a given accuracy. We encode the information of $m$ into $\gamma$ as satisfying

$$(1-\gamma)^m \geq 1 - \xi. \quad (10)$$

In other words, almost all qubits are $|0\rangle$ in this sequence, and $|1\rangle$ appears only after $m$-th digit with probability larger than $1 - \xi$. Owing to this, if $|1\rangle$ appears at the $m'$-th digit for the first time, this is taken as the sign that $m \leq m'$. Based on the observed value $m'$, the length of the output by TM1 (i.e., the presumed length of

the digit of $\beta$) is determined as $n' = \lceil \frac{1}{4} \log_2 m' \rceil$, which ensures

$$\lim_{m'\to\infty} \text{Prob}\left[\left|\frac{N_1(w)}{m'} - \beta\right| < \frac{1}{2^{n'+1}}\right] = 1. \quad (11)$$

With this choice of output length $n'$, guessing $m'$ larger than the true value $m$ does not affect the correctness of the estimation of $\beta$.

### Modification of TM3 in case of thermalization

When we show the undecidability of thermalization, we need to modify TM3 to another TM named TM3+. TM3 flips all A-cells from $a_1$ to $a_2$, and after the flipping TM3 stops. Similarly, TM3+ first flips all A-cells from $a_1$ to $a_2$, but after the flipping TM3+ still runs in order to spend time steps of order $O(L^2)$. Note that TM1 and TM2 take $O(1)$ steps, and TM3 takes $O(L)$ steps. In TM3+, most of the steps before stopping are dominated by those after flipping. This additional trick makes the long-time average of $\mathcal{A}$ with halting TM2 from $\langle e_2|A|e_2\rangle/2$ (in case with TM3) to $\langle e_2|A|e_2\rangle$.

In the construction of TM3+, we introduce two new states of A-cells, $b_l$, and $b_r$, and equip the rule such that the position of $b_l$ is fixed and the position of $b_r$ moves right one cell through a single round trip of the finite control between $b_l$ and $b_r$. At the beginning, $b_r$ sits right of $b_l$, and we set TM3+ stop when $b_r$ hits $b_l$ from left. In this setting, it takes $O(L^2)$ steps until $b_r$ hits $b_l$ from left, which indeed meets the requirement.

### Busy beaver function

The busy beaver function BB($n$) is defined as follows: We consider all possible TMs with $n$ internal states, and start running these TMs with empty inputs. Some TMs will halt, and some other TMs will not. We pay attention only to the former TMs and record the maximum number of steps before halting, which is BB($n$). Since we exclude non-halting TMs, BB($n$) must be finite for all $n$.

We remark that a TM with $m$ internal states and input $u$ with length $l$ can be emulated by another TM with $m + l$ internal states and empty input. The emulation is performed as follows: This TM first outputs the code $u$ on the blank tape by using $l$ internal states, and then works as the TM with $m$ internal states. Conversely, a URTM can emulate any TM with any number of internal states, whose information is given in the input code for the URTM. Thus, the busy beaver function also characterizes the maximum number of steps in terms of the length of the input.

The uncomputability of BB($n$) is a direct consequence of the undecidability of the halting problem. We show this by contradiction. Suppose that BB($n$) is computable for any $n$. Then, for any input code $u$ with length $l$, we run this TM with this input for BB($m + l$) steps and observe whether this TM halts or not. By definition, if this TM does not halt at this step, we can confirm that this TM does not halt forever. This procedure solves the halting problem, which is a contradiction.

The uncomputability of BB(748) is shown by resorting to the fact that there is a TM with 748 internal states such that this TM halts if and only if ZFC is inconsistent[47,48]. Gödel's incompleteness theorem shows that ZFC cannot prove the consistency of ZFC itself if ZFC is consistent. Following a similar argument to above, ZFC cannot compute BB(748) if ZFC is consistent.

**Proof** All the results in this paper are rigorously proved in the Supplementary Note.

## Data availability

Data sharing is not applicable to this article as no datasets were generated or analyzed during the current study.

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

## Acknowledgements

We thank Takahiro Sagawa for stimulating the discussion. N.S. was supported by JSPS Grants-in-Aid for Scientific Research Grant Number JP19K14615.

## Author contributions

N.S. and K.M. contributed equally to this work.

## Competing interests

The authors declare no competing interest.
