## [Peer Review File · Nature Communications]

Reviewers' Comments:

Reviewer #1:

Remarks to the Author:

At the focus of this paper are 1d quantum many-body systems, when the Hamiltonian takes the form of a translation invariant sum of local operators and the initial condition takes the form of a particularly simple product state.

At the heart of the paper is the question whether the long-time average of a given observable is contained within some given (possibly small) interval or not.

The main result is to show by way of specific examples that this question is undecidable in general.

The main problem of the paper seem to me that the main result only applies to the thermodynamic limit, that is, to infinitely large systems:

1. Every real system in nature and on the computer is finite. The present result has no implications for any such system!
2. The thermodynamic limit is essentially meant to be a purely theoretical "trick" in order to make ones life easier. If it turns out to actually make life more complicated, as the present work indicates, one simply should let it be.
3. It seems to me quite reasonable to expect that solving an infinitely large system by means of a computer cannot be done with finite resources. Maybe this is too naive, nevertheless when looking at the present paper from this viewpoint, there remains a suspicion that things may be in some sense quite obvious.
4. As a physicist, it would appear to me more interesting to understand what is going on physically rather than in terms of statements regarding Turing machines: Apparently, upon increasing the systems size, the relaxation process becomes slower and slower. For every finite system, the long-time average is computable, but apparently the problem is that certain limits do not commute here. But how exactly does it happen that one knows the answer to the considered problem for any finite system size but cannot extrapolate from this information what happens in the limit of infinitely large systems?
By the way, there are other systems which may behave similarly, for instance (classical) glasses. So, maybe also for glasses the thermalization problem is undecidable in the sense of the present paper?

In conclusion, this paper appears to me quite provocative and of high originality. However, its importance seems quite a bit oversold.

Reviewer #2:

Remarks to the Author:

In the recent years, many important physical properties have been shown to be algorithmically undecidable: there is no algorithm that, no matter how long it runs, can always decide whether a given property is present in the system. The present manuscript focuses on the problem of whether thermalization in quantum spin systems is an undecidable property.

Most (if not all) of the undecidability results are based on embedding the Halting problem for Turing machines into the physical model considered, in such a way that yes/no instances of the Halting problem correspond to yes/no instances of the property considered. In this work, the authors construct a nearest neighbor spin chain Hamiltonian and a family of initial states, which encodes the input to a Universal Turing Machine (UTM), and show that the expectation value of a certain observable converges to zero or to a positive (computable) value depending on whether the UTM halts on that particular input.

I believe that the problem of understanding which features of physical systems are undecidable is very interesting, some aspect of the current manuscript make me doubt of the impact of the work presented. Specifically, these are some issues I would like the authors to address:

* The definition of thermalization

In order to say that "thermalization is undecidable", one has to make a precise statement about what exactly is the decision problem considered. While the authors claim that they give a precise statement of this result, I am not satisfied by the statement in the Theorem on page 2 (main article).

What are the inputs to the decision problem exactly? In other words, is thermalization a property of the Hamiltonian alone, or of the triple Hamiltonian/input state/observable? From what I understand, the authors can show undecidability if one uses the latter definition. In fact, what would happen if there exists a different pair input state/observable for which one can easily show that the system does *not* thermalize? Could not one then claim already that the system does not thermalize?

If it is indeed the case that the authors consider "thermalization" a property not of the Hamiltonian, but of the triple Hamiltonian/input state/observable (which is the only one that fits their results), then this should be mentioned more explicitly in the paper, and most probably reflected in the abstract itself. This would seem to be a quite weak notion of thermalization.

* The definition of STA and HTA.

The authors do make a more formal definition of two decision problems in the supplementary material, with the definitions of the problems STA and HTA.

In these definition, what is the role of H in STA and ρ in HTA? Are they some sort of "parameters" of the problem, or are they actually inputs to the decision problem? Again, this is important since by reading the definition, one could deduce that the undecidability result holds for any choice of these parameters (up to some restriction such as large enough local Hilbert space dimension), but the authors only show undecidability of the two problems for a single specific choice of H and ρ (respectively).

* The relationship of STA and HTA with the problem of thermalization

The Lemma of the main text and Lemma 1 (supp. mat.) claims that it is undecidable whether the long-term average of A_{L} concentrates near zero or near the expectation value of A under the state e_{2} . In order to connect this to the problem of thermalization of the triple Hamiltonian/input state/observable, one has to connect the long-term average value of

A_L to its thermal average: this seems to me a crucial step, but it is only very briefly brushed upon in the main text ("it can be easily realized" the authors claim in the second paragraph of page 2, without further explanation), and only briefly mentioned in section 10.4 (supp. mat.). I find it unusual that such a crucial point is relegated to a small paragraph at the end of the supplementary material. I suggest the authors expand on this in the main text (referring to the appropriate section of the supplementary material if needed).

My issue with this connection is that I do not see clearly why the thermal average of the observable could not also be by itself uncomputable. In that case, while one cannot decide whether these two quantities are close to zero or strictly positive, one might be able to decide whether their difference is close to zero or not.

In section 10.4 (supp. mat.) it is claimed essentially that one can arbitrarily change the thermal average of the observable without changing its long-term average. First of all, this does not seem to fix the issue of what to do in the case that the thermal average itself is uncomputable (as the change one would need to make is itself uncomputable, the modified observable would become uncomputable). Secondly, if one can arbitrarily change the thermal average of the observable to any desired value, in what sense is the result about thermalization of the same observable? One would believe that there would be physical reasons (whose presence might be as well uncomputable) for which these two quantities coincide, but here we have complete freedom over one of them, and indeed the authors claim that the same observable can be modified to either thermalize or not thermalize. This seems to support my impression that the result is actually weakly connected with thermalization.

On top of the above points, I have some comments about the presentation of the paper which I think should be addressed, as they make the paper less readable and in some points a bit obscure.

* Lemma 1 in supplementary material:

There are a few issues I have found in the statement of this Lemma, and since it is the main result of the manuscript I think there should not be the smallest ambiguity about its content:

- The set $V_{A,\eta}$ is mentioned in Lemma 1 but never actually defined anywhere later in the proof or in the rest of the supplementary material.
- " $V_{A,\eta}$ is large enough to encode all the bit strings" what does this sentence mean? Does it mean $V_{A,\eta}$ is infinite and at least countable?

* Large L limit:

I think the authors should be a bit more explicit in how precisely is the thermodynamic limit (large L limit) taken. Let me explain what I mean.

In the main text, the authors say they consider a spin chain of length L , and then take the limit for L to infinity. They then define a sequence of states, for each value of L , such that the first component (e_0) is different from the rest (e_1) (eq. 1 in main text and eq. 7 in the supplementary material). Usually, when considering the thermodynamic limit of a finite spin chain on an interval, both sides of the interval are sent to infinity (i.e., in the limit the locations of the spin are indexed by integers Z). In this case, the resulting state would be equal to an infinite product of e_1 , since the state e_0 has been "pushed to infinity" on the left. In other words, it would be impossible to find a state on the infinite chain such that its restrictions to finite chains satisfy eq. 7 (supp. material).

This could be fixed by requiring that e_0 is not in the left-most position, but at the origin (the other alternative is to take a one-sided limit, obtaining a system defined only on an half-infinite chain, but I doubt this is preferable). I suspect that the periodic boundary conditions of the Hamiltonian would make the construction work in the same way as before, but I suggest that the authors check this.

* Eq. 8 in main text / eq. 38 in supp. material

This equation, present verbatim in both documents, seems to be missing the dependence of \bar{A} from the initial state ψ (i.e. the coefficients c_i): in fact, as it reads now, it

implies that the long term average of any observable is independent of the initial state, which is clearly absurd. In fact, while this equation is referred to (for example, in section 8.3 and 9.3 of the supplementary material), it is not really used in this form but in its correct form.

Even when corrected, I do not really understand why this (seemingly elementary) statement is considered to be so essential to be present in the main text. I suggest that the authors, after fixing the formula, either explain its importance in the main text or replace it with some more interesting detail about their methodology.

* Eq. 2 in supp. material

The text reads " χ is a characteristic function which takes 1 (resp. 0) if the statement inside the bracket is true (resp. false)", but in the equation χ takes a scalar as an argument.

* The role of M_G

A Turing machine M_G is introduced on page 9 (supp. mat.) but it is not explained what role it plays until page 11. I found it a bit confusing. Maybe the authors should mention what role M_G is going to play in their overview of the proof in Section 5 (supp. mat.).

Reviewer #3:

Remarks to the Author:

Shiraishi and Matsumoto report a technical result that implies that the question of whether a certain class of one-dimensional quantum systems thermalizes is undecidable. I find the result intriguing, and I can well imagine that it may merit publication in Nature Communications. At the same time, the presentation and the level of precision of the writing is such that working through the paper is a painful experience, where I ran into so many instances of unclear statements that in some of the sections I gave up to even try to follow the argument in its full extent. Here comes a non-exhaustive list of some of the points:

(1) In the theorem on page 2 the authors speak of a given(!) observable A , a given value A^* , a given initial state ψ_0 , and a given Hamiltonian H . The statement of the theorem is that than one cannot decide whether thermalization occurs. In this generality, this appears wrong. I certainly can write down an example of (H, A, ψ_0) for which thermalization CAN be proven. Shouldn't the proper statement of the theorem be something like "given (A, ψ_0) , there exist H for which thermalization is undecidable"? (And a similar version with the roles of ψ_0 and H interchanged?) I'm confused!

(2) A lemma, like a theorem, is meant to be an essentially self-contained statement of a result. The Lemma presented at the end of page 2 fails in this respect and is pretty much incomprehensible. Let me go through it sentence by sentence from the perspective of a reader.
--"Set the dimension of the local Hilbert space d to be sufficiently large." I won't be able to do this, because I am given no instructions what "sufficiently large" is. Can this be avoided, as is done in many mathematical proofs, in the form "...then there exists a finite d_0 such that for all local dimensions $d > d_0$ the following holds true:"???
--"Given ... we fix a universal reversible Turing machine (URTM)." Reading this, I must assume that I can fix ANY universal reversible Turing machine that I like, because no specificities are given. This is of course not true. A very specific construction is used. To meet the standards of a lemma, this construction must be described in an equally rigorous way, which should maybe be stated as a separate Lemma, such that it can then be referenced.
--Next the operators V_x are mentioned and x is called the input, but one has no chance to understand where this input comes from, nor how V_x depends on the input. Sure, I can find this out by working through the proofs, but this is not how it is supposed to work. This is frustrating to read and leaves one puzzled.

(Minor remark: the exponent in Eq. (1) must be $^{\otimes(L-1)}$, I presume?)

Short summary: I consider the lemma entirely incomprehensible and hence useless.

(3) Page 3, Classical machines: "TM3 is a simple TM, which flips the state of A-cells if and only if

TM2 halts." This is mentioned just like this, only that A-cells haven't been introduced so far. This may be a minor example, but this is what makes this paper so frustrating to read. Congratulations to anyone who can follow the rest of the section, I can't.

(4) Fig. 2 uses the symbols a_1 , a_2 , $q_{u,r}$, and β , none of which have been introduced. How is one supposed to understand the figure then?

At this point, the confusing and imprecise presentation had discouraged me to an extent that I stopped working through the paper in full detail. I believe it is not the reader's job to decipher things, but the authors' job to explain them as precisely and lucidly as possible. This, in my opinion, Shiraishi and Matsumoto have not achieved to do.

To summarize, I am still intrigued by the paper. I would love to see the paper written in a form such that it can be seriously considered for publication in Nature Communications. Unfortunately I can't recommend the present version to any reader.

Here are a few additional minor remarks:

(5) Page 1: "The ETH claims that all the energy eigenstates of a given Hamiltonian are thermal, that is, indistinguishable from the equilibrium state, as long as we observe macroscopic observables."

(6) Fig. 1a: This illustration may give the incorrect impression that "equilibrium" is one kind of spin configuration (microstate) and "nonequilibrium" is some other kind of spin configuration, whereas these are in fact huge sets of microstates.

(7) Fig. 1b: This illustration seems to suggest that, for equilibration to occur according to the definition of the paper, $\langle A \rangle(t)$ has to remain in a narrow interval around A^* , which is not the case. In fact, the definition of equilibration via the long-time average allows for huge fluctuations of $\langle A \rangle(t)$. Presumably the axis label is just incorrect and should be the long-time average of $\langle A \rangle(t)$, not $\langle A \rangle(t)$ itself.

Response to reviewers

Summary of changes

- We have explicitly stated the undecidability of thermalization as Theorem 2. We have also added the proof idea and its full proof as the section entitled *From relaxation to thermalization* to the main article and Sec. 11 to the Supplementary material. We have added three figures, Figs.5, 6, and 7, for its explanation.
- We have explained the absence of a sufficiently large system size in terms of the busy beaver function in the newly added section entitled *No sufficiently-large system size* in the main article and *Busy beaver function* in the method part. We have also described this point in detail in Sec. 12 of the Supplementary material.
- To clarify the dependency and arbitrariness of several quantities and parameters, we have drastically refined the description around Theorem 1 in the main article, and around Theorem 1a and 1b (Theorem 1 and 2 in the previous manuscript) in the Supplementary material.
- We have added a new section (Sec. 3) to the Supplementary material which provides a brief review of theoretical computer science.
- For readability, we have changed the definition of the generalized URTM denoted by M_G such that M_G also contains TM1. Accordingly, we have modified several description in Sec. 7 and removed the symbol \tilde{M}_G from Sec. 10 of the Supplementary material. In addition, we have modified the meaning of gray regions in Fig. 3.
- To avoid confusion, we have replaced the input code for TM2 from \mathbf{x} to \mathbf{u} .
- To avoid confusion, we have denoted the spatial average of A , $\frac{1}{L} \sum_i A_i$, by calligraphic \mathcal{A} .

Response to Referee 1

Referee 1: The main problem of the paper seem to me that the main result only applies to the thermodynamic limit, that is, to infinitely large systems:

1. Every real system in nature and on the computer is finite. The present result has no implications for any such system!

2. The thermodynamic limit is essentially meant to be a purely theoretical “trick” in order to make ones life easier. If it turns out to actually make life more complicated, as the present work indicates, one simply should let it be.

Since the summary and these two comments concern the same problem, we shall treat them together.

The first comment states that results with thermodynamic limit have no implication for real systems in nature because real systems are finite, and the second comment states that if a complicated result is obtained through the thermodynamic limit, we should not consider it seriously. As far as we see, these two criteria are not supported by standard theoretical physics. In fact, various theoretical rigorous results in statistical mechanics and quantum many-body systems are formulated in the thermodynamic limit. One example is the phase transition in equilibrium statistical mechanics, which is proven to occur only in the thermodynamic limit. Another example is the spectral gap in quantum many-body systems: The presence or absence of the spectral gap (gapful or gapless) can be defined only in the thermodynamic limit. Both the phase transition and the spectral gap are important topics in statistical mechanics and quantum many-body physics.

In addition, some results under the thermodynamic limit show highly complicated phenomena which are worth investigating deeply. The replica symmetry breaking is a complicated type of phase transition, which is one of the central research topics in spin glass physics and information statistical mechanics. We remark that the existence of the replica symmetry breaking in our real three-dimensional world has not yet been shown, and the long-standing debate has continued. Another example is the Haldane conjecture stating qualitative differences between the spectral gap of half-odd-integer spin chains and integer spin chains. This conjecture was first regarded as strange and some physicists do not believe this conjecture, while it is now considered as one of the most important and breakthrough findings in quantum many-body physics. A more striking example is the presence or absence of the spectral gap, which is proven to be undecidable (T. S. Cubitt, *et al.*,

Nature **528**, 207 (2015)). The undecidability of the spectral gap is formulated in the thermodynamic limit and shows an unexpected complicated behavior, while this result is considered to be an important result.

We also comment on the fact that most notions in theoretical computer science, including computational complexity, computability, and the speed of algorithms, are formulated in the large size limit. Since our research is interdisciplinary between physics and theoretical computer science, the large size limit is unavoidable.

On the basis of these facts, we conclude that our result does not decrease its importance by the fact that our result is formulated in the thermodynamic limit.

Referee 1: 3. It seems to me quite reasonable to expect that solving an infinitely large system by means of a computer cannot be done with finite resources. Maybe this is too naive, nevertheless when looking at the present paper from this viewpoint, there remains a suspicion that things may be in some sense quite obvious.

Maybe our previous presentation impresses our results understated. Our claim is that any possible method cannot solve the problem of thermalization, and we mentioned direct numerical simulation just as an intuitive and familiar example which might be inefficient. We would like to clarify two facts:

1. Various problems are solved by indirect approaches efficiently. In contrast to these cases, our result excludes the possibility that the problem of thermalization is solved by any indirect approaches.
2. The problem of thermalization cannot be solved even with *unlimited* computational resource.

We shall clarify the latter point in detail. If our computational resource is large but finite, we cannot compute thermalization in some systems. However, even if our computational resource is unlimited (we can increase the computational power arbitrarily), the problem of thermalization is still out of reach.

In addition, direct numerical simulations usually compute the desired value with a small error if we employ a sufficiently large system size and sufficiently long running time. In the case of near-integrable systems, for example, the Hamiltonian is written as $H = H_{\text{int}} + \varepsilon V$ with integrable Hamiltonian H_{int} and perturbation εV with a small parameter ε , and the necessary system size and time are obtained by the small parameter ε . Thus, with the

unlimited computational resource, we can compute near-integrable systems for any $\varepsilon > 0$. In contrast, the constructed system showing the undecidability of thermalization does not have a sufficiently large system size, and thus even with the unlimited computational resource we cannot compute the fate of thermalization. Therefore, we consider that these behaviors are not obvious ones.

In the revised manuscript, we have added a new section entitled *No sufficiently-large system size* to the main article and a new section (Sec. 12) to the Supplementary material, both of which explain the absence of sufficiently large system size, its background, and unsolvability even with unlimited computational power.

Referee 1: 4. As a physicist, it would appear to me more interesting to understand what is going on physically rather than in terms of statements regarding Turing machines: Apparently, upon increasing the systems size, the relaxation process becomes slower and slower. For every finite system, the long-time average is computable, but apparently the problem is that certain limits do not commute here. But how exactly does it happen that one knows the answer to the considered problem for any finite system size but cannot extrapolate from this information what happens in the limit of infinitely large systems? By the way, there are other systems which may behave similarly, for instance (classical) glasses. So, maybe also for glasses the thermalization problem is undecidable in the sense of the present paper?

Since this comment covers various points, we shall answer them one by one.

In the first sentence, Referee 1 states that they do not prefer statements with Turing machines. I understand that physicists are not familiar with theoretical computer science and Turing machines. However, our study concerns the hardness of problems in physics, which is interdisciplinary research between physics and theoretical computer science. In fact, physics does not have good languages to state the hardness of problems, but theoretical computer science indeed has. Therefore, the use of theoretical computer science is unavoidable. Maybe the question of Referee 1 asks why we use a specific form of a computation system, Turing machines. The reason is that Turing machines can emulate any possible computation system such as C++ and Python, which is declared in the Church-Turing thesis, and thus we can cover all possible computations by just considering Turing machines.

We understand that some readers are not familiar with theoretical computer science. For such readers, we have added a new section (Sec. 3) to the

Supplementary material, which provides a brief review of theoretical computer science.

From the second to fourth sentences, Referee 1 asks what actually happens in the constructed system. In each system, we observe one of “simple thermalization phenomena (the observed value is indeed the long-time average in the thermodynamic limit)” or “extremely long plateau before thermalization (the observed value is in a prethermal plateau far from that in the thermodynamic limit)”, both of which are not so strange at this point. What is highly strange is that we cannot distinguish these two by using any tool. Although we know the full Hamiltonian and we can use unlimited computational resources, we cannot exclude the possibility that the observed stable value is in fact an extremely long prethermal plateau.

To clarify this, we have added the explanation on this point to Sec. 12 of the Supplementary material.

In the last two sentences, Referee 1 comments on the similarity to glassy systems. The connection between glassy dynamics and computational complexity is frequently mentioned (e.g., M. Mezard and A. Montanari, *Information, Physics, and Computation*. Oxford university press (2009)), and thus this comment suggests a very fruitful direction. As far as we see, some systems of glasses always lack thermalization (i.e., the fate of thermalization is decidable), while some marginal systems may leave the fate of thermalization to computational consequences. Thus, although the constructed Hamiltonians have no disorder, they may be related to a subclass of glassy systems. Since the problem of the connection between glassy systems and computational complexity is very deep and no one has a clear answer, we cannot answer the question by Referee 1 at present. In the revised manuscript, we have stated this research direction in the third paragraph of *Discussion* in the main article.

Referee 1: In conclusion, this paper appears to me quite provocative and of high originality.

We thank Referee 1 for finding our paper provocative and highly original.

Referee 1: However, its importance seems quite a bit oversold.

Honest to say, we tried but failed to grasp what point Referee 1 feels as quite a bit oversold. As we have explained, formulation with the thermodynamic limit is a standard framework in theoretical physics, and thus this point does not reduce the importance of our results. Referee 1 may feel

that our paper impresses that further investigation of quantum thermalization is meaningless or any finite-size numerical simulation is unreliable. We fully agree that these impressions are highly overstating and our results do not mean such devastating consequences. To avoid such impressions, we have added a new sentence to the last paragraph of *Discussion* in the main article.

Response to Referee 2

Referee 2: I believe that the problem of understanding which features of physical systems are undecidable is very interesting, some aspect of the current manuscript make me doubt of the impact of the work presented. Specifically, these are some issues I would like the authors to address:

Referee 2: * The definition of thermalization

In order to say that “thermalization is undecidable”, one has to make a precise statement about what exactly is the decision problem considered. While the authors claim that they give a precise statement of this result, I am not satisfied by the statement in the Theorem on page 2 (main article).

What are the inputs to the decision problem exactly? In other words, is thermalization a property of the Hamiltonian alone, or of the triple Hamiltonian/input state/observable? From what I understand, the authors can show undecidability if one uses the latter definition. In fact, what would happen if there exists a different pair input state/observable for which one can easily show that the system does *not* thermalize? Could not one then claim already that the system does not thermalize?

If it is indeed the case that the authors consider “thermalization” a property not of the Hamiltonian, but of the triple Hamiltonian/input state/observable (which is the only one that fits their results), then this should be mentioned more explicitly in the paper, and most probably reflected in the abstract itself. This would seem to be a quite weak notion of thermalization.

We thank Referee 2 for raising two important questions which must be addressed. The first one concerns the input of the decision problem, and the second one concerns the situation that a system with an initial state thermalizes with respect to an observable while the same system with another initial state does not thermalize with respect to another observable.

We first answer the first question. We agree that our previous explanation is a little confused. In the revised manuscript, the input of the decision problem for thermalization is a Hamiltonian alone, and that for relaxation is a Hamiltonian (and a target value A^*). Both the input state and the observable are arbitrarily given and fixed, as far as they satisfy weak conditions (assumptions on the existence of $|\phi_2\rangle$ and $|\phi_3\rangle$). In the proof of undecidability, we construct a proper family of Hamiltonians for any given inputs and observable as far as the conditions are satisfied.

We regret that our previous presentation left this point ambiguous. In the revised manuscript, we have explicitly described this point above Theorem 1 in the main article and in Sec. 4 in the Supplementary material.

We shall proceed to the second question. In our paper, we define thermalization with respect to each observable and with each initial state. Hence, even in the same system (the same Hamiltonian), some initial states may thermalize with respect to some observables and some other initial states may not with respect to some other observables.

One may feel that we should characterize thermalization in an unconditional form, that is, a system is called to thermalize if any initial state thermalizes with respect to any macroscopic observable in this system. Here, we would like to notice the fact that at present no concrete system with local interaction is proven to thermalize in the above sense, and most physicists believe that constructing a thermalizing system is a very hard problem. Since in the proof of undecidability we need to prepare a family of systems where the statement (presence of thermalization) is true, we employ the definition of thermalization not in this unconditional form but in the conditional form with an initial state and an observable. With the latter definition, we can prove thermalization in several systems. In the revised manuscript, we have explicitly stated this point in the fourth paragraph, a newly added paragraph, of *Discussion* in the main article.

Referee 2: * The definition of STA and HTA.

The authors do make a more formal definition of two decision problems in the supplementary material, with the definitions of the problems STA and HTA. In these definition, what is the role of H in STA and ρ in HTA? Are they some sort of “parameters” of the problem, or are they actually inputs to the decision problem? Again, this is important since by reading the definition, one could deduce that the undecidability result holds for any choice of these parameters (up to some restriction such as large enough local Hilbert space dimension), but the authors only show undecidability of the two problems for a single specific choice of H and ρ (respectively).

In the case of STA, the Hamiltonian is fixed to a proper one, but it is independent of the observable A . In the case of HTA, the state is some sort of “parameters” given arbitrarily. In the revised manuscript, we have clarified these points in Sec. 4 in the Supplementary material.

Referee 2: * The relationship of STA and HTA with the problem of thermalization

The Lemma of the main text and Lemma 1 (supp. mat.) claims that it is undecidable whether the long-term average of A_L concentrates near zero or near the expectation value of A under the state e_2 . In order to connect this to the problem of thermalization of the triple Hamiltonian/input state/observable, one has to connect the long-term average value of A_L to its thermal average: this seems to me a crucial step, but it is only very briefly brushed upon in the main text (“it can be easily realized” the authors claim in the second paragraph of page 2, without further explanation), and only briefly mentioned in section 10.4 (supp. mat.). I find it unusual that such a crucial point is relegated to a small paragraph at the end of the supplementary material. I suggests the authors expand on this in the main text (referring to the appropriate section of the supplementary material if needed).

My issue with this connection is that I do not see clearly why the thermal average of the observable could not also be by itself uncomputable. In that case, while one cannot decide whether these two quantities are close to zero or strictly positive, one might be able to decide whether their difference is close to zero or not.

In section 10.4 (supp. mat.) it is claimed essentially that one can arbitrarily change the thermal average of the observable without changing its long-term average. First of all, this does not seem to fix the issue of what to do in the case that the thermal average itself is uncomputable (as the change one would need to make is itself uncomputable, the modified observable would become uncomputable). Secondly, if one can arbitrarily change the thermal average of the observable to any desired value, in what sense is the result about thermalization of the same observable? One would believe that there would be physical reasons (whose presence might be as well uncomputable) for which these two quantities coincide, but here we have complete freedom over one of them, and indeed the authors claim that the same observable can be modified to either thermalize or not thermalize. This seems to support my impression that the result is actually weakly connected with thermalization.

We agree with the comments of Referee 2 in two points: First, our previous explanation is very ambiguous and lacks rigor, and second, the previous form of our result on the undecidability of thermalization is a weak result. Triggered by the latter comment, we tried to prove a stronger form of undecidability, and fortunately succeeded in proving it. Now our theorem (Theorem 2 in the revised manuscript) on thermalization takes the form that both the initial state and the observable are arbitrarily given, and only the Hamiltonian is the input of the decision problem. The key idea in this im-

provement is to change the orthonormal basis $\{|e_i\rangle\}$ instead of the observable A .

We have added a new section entitled *From relaxation to thermalization* to the main article, and a new section (Sec. 11) to the Supplementary material in order to demonstrate the undecidability of thermalization.

Referee 2: On top of the above points, I have some comments about the presentation of the paper which I think should be addressed, as they make the paper less readable and in some points a bit obscure.

Referee 2: * Lemma 1 in supplementary material:

There are a few issues I have found in the statement of this Lemma, and since it is the main result of the manuscript I think there should not be the smallest ambiguity about its content:

- The set $\mathcal{V}_{A,\eta}$ is mentioned in Lemma 1 but never actually defined anywhere later in the proof or in the rest of the supplementary material.

- “ $\mathcal{V}_{A,\eta}$ is large enough to encode all the bit strings” what does this sentence mean? Does it mean $\mathcal{V}_{A,\eta}$ is infinite and at least countable?

What we intended to state in “ $\mathcal{V}_{A,\eta}$ is large enough to encode all the bit strings” is that $\mathcal{V}_{A,\eta}$ has infinitely many unitary operators and we have a proper correspondence between any bit string to a unitary operator. However, as commented by Referee 2, the symbol $\mathcal{V}_{A,\eta}$ does not appear after the Lemma. Therefore, we have decided not to use the symbol $\mathcal{V}_{A,\eta}$.

Referee 2: * Large L limit:

I think the authors should be a bit more explicit in how precisely is the thermodynamic limit (large L limit) taken. Let me explain what I mean.

In the main text, the authors say they consider a spin chain of length L , and then take the limit for L to infinity. They then define a sequence of states, for each value of L , such that the first component (e_0) is different from the rest (e_1) (eq. 1 in main text and eq. 7 in the supplementary material). Usually, when considering the thermodynamic limit of a finite spin chain on an interval, both sides of the interval are sent to infinity (i.e., in the limit the locations of the spin are indexed by integers \mathbb{Z}). In this case, the resulting state would be equal to an infinite product of e_1 , since the state e_0 has been “pushed to infinity” on the left. In other words, it would be impossible to find a state on the infinite chain such that its restrictions to finite chains satisfy eq. 7 (supp. material).

This could be fixed by requiring that e_0 is not in the left-most position, but at the origin (the other alternative is to take a one-sided limit, obtaining

a system defined only on an half-infinite chain, but I doubt this is preferable). I suspect that the periodic boundary conditions of the Hamiltonian would make the construction work in the same way as before, but I suggest that the authors check this.

As guessed by Referee 2, the periodic boundary condition and the almost uniform initial state ($|\phi_0\rangle \otimes |\phi_1\rangle^{\otimes L-1}$) makes our argument consistent. Owing to the translation invariance of the Hamiltonian and the observable, the initial state with $1 \leq i \leq 2L$ and that with $-L + 1 \leq i \leq L$ provide the same physics by just relabeling the sites $-L + 1 \leq i \leq 0$ in the latter system as $j = 2L + i$. We have added a comment on this point as footnote 5 in the Supplementary material.

Referee 2: * Eq. 8 in main text / eq. 38 in supp. material

This equation, present verbatim in both documents, seems to be missing the dependence of \bar{A} from the initial state ψ (i.e. the coefficients c_i): in fact, as it reads now, it implies that the long term average of any observable is independent of the initial state, which is clearly absurd. In fact, while this equation is referred to (for example, in section 8.3 and 9.3 of the supplementary material), it is not really used in this form but in its correct form.

Even when corrected, I do not really understand why this (seemingly elementary) statement is considered to be so essential to be present in the main text. I suggest that the authors, after fixing the formula, either explain its importance in the main text or replace it with some more interesting detail about their methodology.

We thank Referee 2 for pointing out the typo and suggestions. Following their advice, we have fixed the typo and removed this explanation from the main text.

Referee 2: * Eq. 2 in supp. material

The text reads " χ is a characteristic function which takes 1 (resp. 0) if the statement inside the bracket is true (resp. false)", but in the equation χ takes a scalar as an argument.

In Eq.(2) in the previous manuscript, the argument of χ is the statement " $\langle \psi(t) | \mathcal{A} | \psi(t) \rangle \simeq \text{Tr}[\mathcal{A} \rho^{\text{MC}}]$ ", which is not a scalar but a relation. At the same time, we noticed that many brackets in Eq.(2) decrease readability, and in addition, the above statement left some ambiguity. Therefore, in the revised manuscript we have modified Eq.(2) as

$$\lim_{V \rightarrow \infty} \lim_{T \rightarrow \infty} \frac{1}{T} \int_0^T dt \chi\{|\langle \psi(t) | \mathcal{A} | \psi(t) \rangle - \text{Tr}[\mathcal{A} \rho^{\text{MC}}]|\} < \varepsilon\} = 1 \quad (1)$$

Referee 2: * The role of M_G

A Turing machine M_G is introduced on page 9 (supp. mat.) but it is not explained what role it plays until page 11. I found it a bit confusing. Maybe the authors should mention what role M_G is going to play in their overview of the proof in Section 5 (supp. mat.).

Following their advice, we have clarified why we generalize the URTM, which is stated in Sec. 6 and Sec. 7.2 in the revised Supplementary material.

Response to Referee 3

Referee 3: Shiraishi and Matsumoto report a technical result that implies that the question of whether a certain class of one-dimensional quantum systems thermalizes is undecidable. I find the result intriguing, and I can well imagine that it may merit publication in Nature Communications. At the same time, the presentation and the level of precision of the writing is such that working through the paper is a painful experience, where I ran into so many instances of unclear statements that in some of the sections I gave up to even try to follow the argument in its full extent. Here comes a non-exhaustive list of some of the points:

Firstly, we apologize for our poor presentation. Below, we answer all the ambiguous points raised by Referee 3.

Referee 3: (1) In the theorem on page 2 the authors speak of a given(!) observable A , a given value A^* , a given initial state ψ_0 , and a given Hamiltonian H . The statement of the theorem is that than one cannot decide whether thermalization occurs. In this generality, this appears wrong. I certainly can write down an example of (H, A, ψ_0) for which thermalization CAN be proven. Shouldn't the proper statement of the theorem be something like "given (A, ψ_0) , there exist H for which thermalization is undecidable"? (And a similar version with the roles of ψ_0 and H interchanged?) I'm confused!

The structure of the decision problem on relaxation (*Theorem* in the previous manuscript, and *Theorem 1* in the revised manuscript) is as follows (Here we dropped the amount of errors and some conditions on the initial states, observables, and Hamiltonians to avoid confusion.):

1. The observable A and the initial state $|\psi_0\rangle$ are given arbitrarily. These two are fixed parameters of the decision problem.
2. Our task is to decide whether this observable from this initial state relaxes to a given target value A^* under a given Hamiltonian H . The Hamiltonian H and the target value A^* are inputs for this decision problem, and we need to solve the above decision problem for all H and A^* .

If there exists a procedure to solve this decision problem for all H and A^* , this decision problem is decidable. If there is no procedure to solve this decision problem for all H and A^* , this decision problem is undecidable. To prove

undecidability, it suffices to show that this decision problem is undecidable for a particular form of H and A^* . This is what we did in our manuscript.

We regret that our previous manuscript left this important point ambiguous. In the revised manuscript, we have explicitly stated these points above Theorem 1 in the main article and in Sec. 4 in the Supplementary material.

Referee 3: (2) A lemma, like a theorem, is meant to be an essentially self-contained statement of a result. The Lemma presented at the end of page 2 fails in this respect and is pretty much incomprehensible. Let me go through it sentence by sentence from the perspective of a reader.

–"Set the dimension of the local Hilbert space d to be sufficiently large." I won't be able to do this, because I am given no instructions what "sufficiently large" is. Can this be avoided, as is done in many mathematical proofs, in the form "...then there exists a finite d_0 such that for all local dimensions $d > d_0$ the following holds true:"???

We agree that our previous presentation on the dimension of the local Hilbert space is unclear. However, at the same time, if we employ "there exists \sim such that" syntax, the statement of the lemma contains so many sentences in this type, which also makes our manuscript unreadable. Since the size of a dimension is not so important in our context, we have moved the description on the dimension to the beginning of *Statement of main results* in the main article.

Referee 3: –"Given ... we fix a universal reversible Turing machine (URTM)." Reading this, I must assume that I can fix ANY universal reversible Turing machine that I like, because no specificities are given. This is of course not true. A very specific construction is used. To meet the standards of a lemma, this construction must be described in an equally rigorous way, which should maybe be stated as a separate Lemma, such that it can then be referenced.

Honest to say, we are a little confused by this comment. In our proof, for a given URTM on a 01 bit tape, we construct a classical TM (called *generalized RTM* in the Supplementary material) which emulates any given URTM as TM2. The generalized RTM, which consists of TM1, TM2, and TM3, takes a specific construction, while we can implement any URTM in this generalized RTM as TM2.

In the revised manuscript, we have clarified the difference between the implemented URTM (which is arbitrary) and the generalized RTM (which

employs a specific form) above Lemma and the beginning of *Classical machines* in the main article.

Referee 3: –Next the operators V_x are mentioned and x is called the input, but one has no chance to understand where this input comes from, nor how V_x depends on the input. Sure, I can find this out by working through the proofs, but this is not how it is supposed to work. This is frustrating to read and leaves one puzzled.

We thank Referee 3 for pointing out our insufficient explanation. We regret that we introduce V_x before introducing the input code x . (In the revised manuscript, the symbol for the input code is changed to \mathbf{u} .) In the revised manuscript, we have modified the description.

Referee 3: (Minor remark: the exponent in Eq. (1) must be $\otimes(L - 1)$, I presume?)

As correctly presumed by Referee 3, it is our typo. We have fixed it.

Referee 3: Short summary: I consider the lemma entirely incomprehensible and hence useless.

We hope that the revised version of Lemma now becomes readable.

Referee 3: (3) Page 3, Classical machines: "TM3 is a simple TM, which flips the state of A-cells if and only if TM2 halts." This is mentioned just like this, only that A-cells haven't been introduced so far. This may be a minor example, but this is what makes this paper so frustrating to read. Congratulations to anyone who can follow the rest of the section, I can't.

We again regret that we used the term *A-cells* before introducing it. In the revised manuscript, we introduce A-cells and M-cells at the beginning of the section entitled *Classical machines*.

Referee 3: (4) Fig. 2 uses the symbols a_1 , a_2 , q_u , r , and β , none of which have been introduced. How is one supposed to understand the figure then?

In the revised manuscript, we have put the explanation on a_1 , a_2 , and β in the main text and in the caption of Fig.2. Since q_u is an example of the internal state of TM2, we have explained q_u (and r , which is the unique internal state of TM3) only in the caption of Fig.2.

Referee 3: At this point, the confusing and imprecise presentation had discouraged me to an extent that I stopped working through the paper in full detail. I believe it is not the reader's job to decipher things, but the authors' job to explain them as precisely and lucidly as possible. This, in my opinion, Shiraishi and Matsumoto have not achieved to do.

To summarize, I am still intrigued by the paper. I would love to see the paper written in a form such that it can be seriously considered for publication in Nature Communications. Unfortunately I can't recommend the present version to any reader.

We apologize for our insufficient and unreadable explanation. At the same time, we are grateful that Referee 3 sees the potential value of our results despite our bad presentation.

Referee 3: (5) Page 1: "The ETH claims that all the energy eigenstates of a given Hamiltonian are thermal, that is, indistinguishable from the equilibrium state, as long as we observe macroscopic observables."

We are afraid that Referee 3 forgot to write a comment on this sentence.

Referee 3: (6) Fig. 1a: This illustration may give the incorrect impression that "equilibrium" is one kind of spin configuration (microstate) and "nonequilibrium" is some other kind of spin configuration, whereas these are in fact huge sets of microstates.

To avoid this confusion, following their advice, we draw two microscopic states corresponding to equilibrium and nonequilibrium, which suggests the existence of other various numbers of equilibrium and nonequilibrium microstates.

Referee 3: (7) Fig. 1b: This illustration seems to suggest that, for equilibration to occur according to the definition of the paper, $\langle A \rangle(t)$ has to remain in a narrow interval around A^* , which is not the case. In fact, the definition of equilibration via the long-time average allows for huge fluctuations of $\langle A \rangle(t)$. Presumably the axis label is just incorrect and should be the long-time average of $\langle A \rangle(t)$, not $\langle A \rangle(t)$ itself.

What we actually plot is an instantaneous value $\langle \psi(t) | A_L | \psi(t) \rangle$ at time t , where A_L is the density of A : $A_L := \frac{1}{L} \sum_{i=1}^L A_i$. The density has no fluctuation in the thermodynamic limit.

At the same time, we understand that it is highly confusing to use the same symbol A to represent both an observable on a single site and its spatial average. To avoid this confusion, we denote the spatial average of A by calligraphy \mathcal{A} .

Reviewers' Comments:

Reviewer #1:

Remarks to the Author:

With the revised version of their work, the authors succeeded to overcome my main concerns.

Also in view of the other reports, I must conclude that the presentation of the original paper was rather poor. My own mistake was to think that the authors take the thermodynamic limit first, and only then consider the long time average of the already infinitely large system. In fact, this misunderstanding is still suggested by the key Fig. 1b, and also at some places in the text.

But since the paper actually considers the long-time limit first and only then takes the thermodynamic limit, I agree that the obtained results are of high interest.

I still feel that the presentation is quite far from optimal. But I do not consider it to be my duty to take the quite substantial effort which is probably necessary to significantly improve the situation.

In conclusion, I think that content wise the paper is now recommendable for publication in Nature Communications, while the presentation still amounts to a borderline case.

Reviewer #2:

Remarks to the Author:

The authors have addressed the issues I had raised in a very satisfactory manner. I think the quality of the exposition after the first round of review has increased noticeably, and the ambiguous definitions have been corrected.

In particular, I had some very fundamental questions about the impact of their result on the problem of thermalization, based on 1. the possibility of modifying the thermal average of the observable and 2. the need of fixing both initial state and observable for their result to hold. Both of these are now discussed very nicely in the paper, and my objections no longer apply.

I recommend the paper to the published.

Response to reviewers

Summary of changes

- We have explicitly stated the fact that we first take the long-time limit and then take the thermodynamic limit in both the first paragraph of *Statement of the main result* and the legend of Fig.1.
- We have modified the symbol in Fig.1(b) from $\langle\psi(t)|\mathcal{A}|\psi(t)\rangle$ to $\langle\psi(t)|\mathcal{A}_L|\psi(t)\rangle$ in order to avoid confusion.
- We have added the section of *Data Availability*.
- We have removed some italics and underlines.

Response to Referee 1

Referee 1: With the revised version of their work, the authors succeeded to overcome my main concerns.

Also in view of the other reports, I must conclude that the presentation of the original paper was rather poor. My own mistake was to think that the authors take the thermodynamic limit first, and only then consider the long time average of the already infinitely large system. In fact, this misunderstanding is still suggested by the key Fig. 1b, and also at some places in the text.

But since the paper actually considers the long-time limit first and only then takes the thermodynamic limit, I agree that the obtained results are of high interest.

I still feel that the presentation is quite far from optimal. But I do not consider it to be my duty to take the quite substantial effort which is probably necessary to significantly improve the situation.

In conclusion, I think that content wise the paper is now recommendable for publication in Nature Communications, while the presentation still amounts to a borderline case.

We sincerely appreciate Referee 1's recommendation to publish our manuscript in Nature Communications.

At the same time, we notice what point Referee 1 was confused. To avoid such confusion, in the revised manuscript we have explained explicitly the fact that we first take the long-time limit and then take the thermodynamic limit, to the bottom of the first paragraph of *Statement of the results* and the legend of Fig.1(b). We have also modified the symbol in Fig.1(b) from $\langle \psi(t) | \mathcal{A} | \psi(t) \rangle$ to $\langle \psi(t) | \mathcal{A}_L | \psi(t) \rangle$.

Response to Referee 2

Referee 2: The authors have addressed the issues I had raised in a very satisfactory manner. I think the quality of the exposition after the first round of review has increased noticeably, and the ambiguous definitions have been corrected.

In particular, I had some very fundamental questions about the impact of their result on the problem of thermalization, based on 1. the possibility of modifying the thermal average of the observable and 2. the need of fixing both initial state and observable for their result to hold. Both of these are now discussed very nicely in the paper, and my objections no longer apply.

I recommend the paper to the published.

We thank Referee 2 for recommending our manuscript to publish.